# Sex Differences in Patterns of Childhood Traumatic Experiences in Chinese Rural-To-Urban Migrant Children

**DOI:** 10.3390/children10040734

**Published:** 2023-04-16

**Authors:** Yiming Liang, Ruiyao Wu, Qi Huang, Zhengkui Liu

**Affiliations:** 1Shanghai Key Laboratory of Mental Health and Psychological Crisis Intervention, Affiliated Mental Health Center (ECNU), School of Psychology and Cognitive Science, East China Normal University, Shanghai 200062, China; 2CAS Key Laboratory of Mental Health, Institute of Psychology, Chinese Academy of Sciences, Beijing 100101, China

**Keywords:** childhood traumatic experiences, sex differences, latent class analysis, migrant children, Chinese

## Abstract

Background: Children and adolescents are likely to be exposed to various types of childhood traumatic experiences (CTEs) with gender-specific patterns. Rural-to-urban migrant children have been demonstrated a greater risk of CTE exposure than local children. However, no study has investigated sex differences in the patterns of CTEs and predictive factors among Chinese children. Methods: A large-scale questionnaire survey of rural-to-urban migrant children (N = 16,140) was conducted among primary and junior high schools in Beijing. Childhood trauma history, including interpersonal violence, vicarious trauma, accidents and injuries was measured. Demographic variables and social support were also examined. Latent class analysis (LCA) was utilized to examine patterns of childhood trauma, and logistic regression was used to examine predictors. Results: Four classes of CTEs were found among both boys and girls, labeled low trauma exposure, vicarious trauma exposure, domestic violence exposure, and multiple trauma exposure. The possibility of various CTEs in the four CTE patterns was higher among boys than girls. Sex differences also manifested in predictors of childhood trauma patterns. Conclusions: Our findings shed light on sex differences in CTE patterns and predictive factors in Chinese rural-to-urban migrant children, suggesting that trauma history should be considered along with sex, and sex-specific prevention and treatment programs should be developed.

## 1. Introduction

The large-scale rural-to-urban migration in China has been a unique social phenomenon following the acceleration of urbanization [1]. As a result, millions of migrant children are moving to large cities with their parents. According to Chinese National Bureau of Statistics, it was estimated that approximately 34 million migrant children lived in urban China, with the majority of them being school-age children. Rural-to-urban migrant children in China suffer from a range of migration-related disadvantages compared with their local peers, including economic challenges, educational inequality, and social discrimination [2,3]. The dualistic household registration system that assigns the children to either rural or urban status limits their access to social welfare and educational resources, making them vulnerable to discrimination based on their migrant status [4]. Furthermore, the marginalized living conditions experienced by many migrant children also lead to reduced parental supervision, which can result in poor parent–child attachments [5]. In line with this, they are often exposed to childhood traumatic experiences (CTEs), which are linked to a series of negative mental, physical and behavioral outcomes [6,7,8,9]. For example, they experienced more community violence, punishment from parents, and negative life events than their urban peers [10,11,12]. Therefore, to prevent potential exposure to CTEs and shed light on possible interventions and policies for them, research is needed to gain a deeper understanding of the unique experience of this group of children in China. In addition, gender has been a crucial factor associated with patterns of CTEs [13,14,15,16]. Thus, it is crucial to explore sex differences in CTEs and predictive factors of CTE patterns, providing further information for gender-specific interventions for migrant boys and girls.

Researchers have recognized that exposure to CTEs tends to include a pattern of CTE exposure, instead of being limited to single CTEs [7,16]. It has been demonstrated that different CTE types often cooccur [16,17], and exposure to one type of CTE significantly increases the risk for additional CTEs [18]. To reflect the patterns of such co-occurrence, an increasing number of studies on child development have recently adopted person-centered methodological approaches, such as latent class analysis (LCA), to examine the heterogeneity of CTEs [19]. An LCA suggests that a sample may be divided into several mutually exclusive and exhaustive subgroups named classes [20]. The individuals from one latent class are characterized by similar responses on a series of variables, demonstrating a pattern of responses different from other classes. Studies on CTE patterns with person-centered approaches have yielded preliminary results for a more accurate description of trauma histories and better insights into specific psychological consequences related to different patterns [6,16,21].

Prior LCA research has discovered two to six groups with distinct trauma profiles in samples of different characteristics [16,21]. Specific patterns of CTE exposure were observed among children [22], as most researchers revealed four classes of childhood trauma in adolescents, often including a class with a low possibility of all CTEs and a class with multiple trauma exposures [16,23,24]. A study using a nationally representative sample of 10,123 US adolescents found four classes: low-risk (79%), combined interpersonal nonsexual and sexual trauma (4.6%), interpersonal nonsexual trauma (15.6%), and high exposures for all trauma classes (1.3%) [24]. Other numbers of classes also emerged due to differences in culture, samples, and measurement [6,14]. For example, three classes of CTEs were supported by data from adolescents and young adults in an American urban community sample, with a predominant-female class characterized by exposure to sexual assaults, a predominantly male class characterized by violence exposure, and a final class with low levels of CTEs [6]. Therefore, the current study adopted LCA to examine subgroups of migrant children exhibiting different patterns of CTE cooccurrence. This approach could compare the cooccurrence of CTEs between the male and female samples of migrant children.

With increasing recognition of the existence of particular patterns of CTEs, researchers have paid closer attention to potential sex differences. Notable sex differences were observed in experiencing different CTE types, with consistent results of males being more frequently exposed to physical abuse and females reporting higher exposure to sexual abuse or crime [14,15,25,26,27]. In addition, some research suggested that males had greater odds of being exposed to accidents and witnessing death or injury, whereas females had a higher risk in experiencing intimate partner violence [28,29].

Moreover, previous studies showed that CTE patterns in males and females differed across cultures [13,16,30]. In a Danish sample, configurations of childhood adversity for girls were more complex and diverse, with five latent classes among girls and three classes among boys [14]. The high poly-adversity class and sexual abuse class only emerged from the female sample, and the majority of girls fell into several multiple exposed subgroups characterized by combined sexual abuse, physical abuse and other adversities. Most boys were in classes characterized by physical abuse and other adversities. However, a study on US adults identified three distinct classes of childhood adversities in males and four in females, with an additional class for females to explain the co-occurrence of sexual and physical abuse [15]. Males in the sexual abuse class had a higher probability of experiencing sexual abuse and molestation than females in the same class. Nonetheless, most trauma studies on sex differences in childhood trauma patterns only dealt with Western samples, requiring more evidence from different cultural backgrounds.

To tailor specific intervention measures and psychological work for boys and girls, environmental and personal risk factors for CTE types need to be identified and examined. Familial factors such as parents’ low socioeconomic status (SES; e.g., low education levels) and inadequate child supervision (e.g., a child living apart from biological parents) were associated with a higher risk of being exposed to multiple CTEs [15,31]. Children with multiple trauma exposures were significantly more likely to come from single-parent families or stepfamilies [17]. In a latent class analysis on CTEs of US adolescents, compared to the class with low risk of trauma exposure, those in the other three classes labeled as the sexual assault risk class, nonsexual risk class and high risk class were considerably less likely to live with any one of their biological parents [24]. In addition to family factors, peer support is an important protective factor for children and adolescents [32]. In terms of personal risk factors, older age was a recognized risk factor of CTEs, probably because it entailed more time to accumulate CTEs [33,34].

To date, little is known about sex differences in patterns of CTEs and predictive factors of CTE patterns among Chinese rural-to-urban migrant children. The present study addressed the gap by (1) assessing sex-specific patterns of CTEs and (2) comparing environmental and personal predictive factors of CTE patterns of boys and girls in a large-scale sample of migrant children who moved from rural to urban areas in Beijing.

## 2. Methods

### 2.1. Procedure and Participants

This study utilized data from an extensive survey on migrant children who had relocated from rural areas to urban areas of Beijing. The sample came from 58 primary or middle schools which were set up primarily for migrant children. These schools were located around the border of rural and urban areas, where the migrant workers often concentrate, had registered with Beijing Education Bureau, and enrolled more than 200 students. 

Prior informed consent was obtained from the children in a written form and clearly informed that they had the right to refuse to answer any question or withdraw their participation at any point. Oral informed consent was obtained from their parents or guardians. Furthermore, the study procedure was approved by the school representatives and administrators before being conducted. 

The children were measured collectively during class time, guided by two assistant investigators who were trained to distribute questionnaires and give standardized instructions. The study design and procedures were approved by the ethics review committee of the Institute of Psychology, Chinese Academy of Sciences (Protocol number: H13024). A full description of school selection, sampling procedures, and data collection is available elsewhere [23,35].

The original sample was composed of 16,682 participants, and 16,140 (96.75%) questionnaires were usable after excluding incomplete or inaccurate responses. Multiple imputation was used to handle the item-level missing values (2.41%). Of the final sample, 57.3% were male, 41.3% were female, and their ages ranged from 8 to 17 years. Their geographical distribution covered 29 regions in China’s mainland, representing major outflow areas of migrant children. Among them, 31 children were born in Beijing, and some had moved to Beijing with their parents when they were very young, but they could not attend public schools with the local children due to the lack of Beijing hukou. The sociodemographic features of the sample are presented in Table 1. A relatively large percentage of data was missing regarding the child’s age when parents migrated for work, possibly because only one of the parents migrated for work, or they left the hometown when the children was too young to remember.

### 2.2. Measures

#### 2.2.1. Demographic Information

A questionnaire was administered to collect demographic characteristics, such as age, gender, parents’ education level, and parents’ marital status. Family support was measured by choosing the number of close family members they had, and peer support was measured by choosing the number of close friends they had. The options for both items included “1 = none, 2 = one or two, 3 = three to five, 4 = six and above”.

#### 2.2.2. Traumatic Experience History

Respondents were measured on the lifetime occurrence of 11 traumatic events with the first section of the University of California at Los Angeles (UCLA) PTSD reaction index for the DSM-IV, revision 1 (UCLA PTSD-RI) [36]. The checklist screens for 12 types of traumatic events which meet the DSM-IV A1 criteria, including exposure to natural disasters, medical trauma, community violence, and domestic violence. Among them, the item about war was removed as all participants had no experience of it. Therefore, 11 types of CTEs were measured as dummy-coded variables (0 = absent; 1 = present) for the study of CTE patterns among migrant children. In line with the classification used in previous research [31,33], these events were classed into three groups: accidents and injuries (4 items), interpersonal violence (3 items), and trauma within the social network or witnessing events (4 items).

### 2.3. Data Analysis

Then, latent class analysis (LCA) was conducted to identify patterns of childhood trauma across the 11 binary items of CTEs. Two sets of models were examined for boys and girls separately. Models with 1–5 classes were estimated using Mplus version 8.2 [37].

To choose the optimal number of latent classes for the male and female model, fit statistics were examined and compared using the Bayesian information criterion (BIC), sample-size-adjusted BIC (a-BIC), Akaike information criterion (AIC) indices, and the Lo-Mendell–Rubin likelihood ratio test (LMR–LRT) value. Among the set of models for males, those with lower AIC, BIC, and a-BIC values were considered the best fitting [38,39]. The LMR–LRT compares a solution with k classes to the one with k − 1 classes, whereby a significant result indicates that the model with k classes fits the sample better [40]. The same applied to the models for females. In addition, interpretability was considered in class selection. The smallest class needed to make up no less than 5% to ensure the scalability of LCA results [39].

Next, the potential predictors of CTE classes were determined on the basis of theoretical and statistical considerations. A number of variables supported by previous literature [15,30,31] were included in Chi-squared tests among the classes of CTEs, such as age, gender, parental marital status, parents’ education level, family support, and peer support. The predictors that had significant results were entered into the following logistic regression analysis.

Lastly, we investigated the sex differences in potential predictors of CTE classes. Two sets of multinomial logistic regression of the latent class variables with the demographic variables were performed with the male and female samples separately, and missing data in the independent variables were replaced by multiple imputation [41].

## 3. Results

### 3.1. Classes of Childhood Trauma

Table 2 displays the fit indices of the two- to five-class models for male and female participants. For males, the model with four classes fit the data better, with the lowest BIC and a significant value of LMR–LRT. Although the AIC and adjusted BIC continued to decrease in the five-class model, the percentage of the smallest class dropped to only 0.5%, indicating low scalability. Thus, the model with four classes was chosen as the final model for males.

Similarly, a four-class model was considered as the best one for females, with the lowest BIC and adjusted BIC values. In addition, the LMR–LRT indicated that the model with five classes was not significantly better than the one with four classes, despite the slight reduction in AIC in the five-class model. Thus, the four-class solution was maintained for females as well.

The characteristics of the four classes in male and female participants are depicted in Figure 1. Similar class patterns emerged from the male and female samples. Specifically, the following four classes were identified for both genders: (a) low trauma exposure class (57.2% in males; 64.6% in females), which showed no or extremely low probability of endorsing any type of CTE; (b) multiple trauma exposure class (5.4% in males; 5.3% in females), which showed moderate or high probability across all CTE types; (c) domestic violence exposure class (9.7% in males; 13.0% in females), which showed a moderate possibility of seeing traumatic events and a strong likelihood of experiencing or seeing domestic violence; (d) vicarious trauma exposure class (27.7% in males; 17.2% in females), which demonstrated a moderate likelihood of seeing traumatic events or experiencing painful medical treatment.

However, there were sex differences in terms of the shape and proportion of childhood trauma patterns. Although the low trauma exposure class was the largest group for both genders, the proportion of males belonging to this class (57.2%) was lower than that of females (64.6%). Meanwhile, compared to females, males in the low trauma exposure class had a higher probability of some CTE types, including witnessing community violence and seeing a dead body.

Although similar proportions of males and females fell into the multiple trauma exposure class (5.4% and 5.3%), almost all possible CTE types in males were higher than those in females. Specifically, the likelihood of seeing and experiencing violence at home or away from home, along with sexual harassment, was especially higher for males in this class.

The proportion of the domestic violence exposure class was larger in females (13.0%) than in males (9.7%). Except for witnessing violence away from home, the probability of almost all other types of CTE was generally higher for males than females in this class, which indicated that males in the domestic violence exposure class were more likely to suffer from domestic violence and see violence at home or away from home than females in the same class.

The vicarious trauma exposure class represented more male participants (27.7%) than female participants (17.2%). Males in this class demonstrated a higher possibility of most CTE types, especially experiencing painful medical treatment and domestic violence.

### 3.2. Predictors of CTE Classes for Boys and Girls

Table 3 and Table 4 show the results of χ^2^ tests performed on demographic variables and potential predictors across the four CTE classes. For both genders, the childhood trauma class was significantly related to age, family support, peer support, father’s education and parental marital status. To further explore sex differences in the predictors of trauma classes, multiple logistic regression was conducted separately in the boys and girls.

The results of multiple logistic regression for males and females are presented in Table 5 and Table 6. Several sex differences existed in the prediction of classes based on parental marital status, peer support, and father’s education level.

In general, females with remarried parents and males with divorced parents had a higher likelihood of being in classes with more CTE types. Specifically, among male participants, those with divorced parents had higher odds of falling into the domestic violence exposure class (domestic vs. low (latter is reference group, same below): odds ratio [OR] = 1.80, 95% confidence interval [CI] = [1.22, 2.65], *p* < 0.01) and multiple trauma exposure class (multiple vs. low: OR = 2.16, 95% CI = [1.37, 3.40], *p* < 0.001), compared to those with parents in their first marriage. Similarly, females with divorced parents had significantly higher odds of falling into the multiple trauma exposure class (multiple vs. low: OR = 2.82, 95% CI = [1.67, 4.74], *p* < 0.001). Moreover, females with remarried parents had a greater chance of being categorized into the domestic violence exposure class (domestic vs. low: OR = 1.66, 95% CI = [1.22, 2.27], *p* < 0.01) and vicarious trauma exposure class (vicarious vs. low: OR = 1.42, 95% CI = [1.20, 1.98], *p* < 0.05) than females whose parents were in their first marriage.

Fathers’ education level predicted different classes for males and females. A primary school education level or below in fathers was predictive of the domestic violence exposure class in males (domestic vs. low: OR = 1.34, 95% CI = [1.08, 1.65], *p* < 0.01) and the multiple trauma exposure class in females (multiple vs. low: OR = 1.50, 95% CI = [1.04, 2.16], *p* < 0.05). In addition, peer support also differentiated between different classes for males and females. A higher level of peer support decreased the likelihood of falling in the multiple trauma exposure class for males (multiple vs. low: OR = 0.88, 95% CI = [0.77, 0.99], *p* < 0.05) and the domestic violence exposure class for females (domestic vs. low: OR = 0.86, 95% CI = [0.78, 0.96], *p* < 0.01).

Few sex differences were observed in relation to children’s age and family support, as they were predictive of trauma classes for both genders. Generally, being of an older age was a risk factor, which related to classes having more CTE types. Family support differentiated several classes significantly for both genders. A lower level of family support was related to the domestic violence exposure class and multiple trauma exposure class.

## 4. Discussion

The current study extends previous trauma research by providing the first investigation on CTE patterns of rural-to-urban Chinese migrant boys and girls and exploring predictive factors of CTE classes through a sex-specific lens. Three main findings emerged from the current study: (1) both genders were characterized by four classes of CTEs that were similar in nature, namely, a low trauma exposure class, vicarious trauma exposure class, domestic violence exposure class, and multiple trauma exposure class; (2) sex differences were observed from the patterns of four classes of CTEs; (3) predictive factors such as parents’ marital status, fathers’ education level and peer support were associated with different CTE class membership among boys and girls.

### 4.1. Sex Differences in CTE Patterns

Four CTE patterns were found in both sexes, characterized by low trauma exposure class, vicarious trauma exposure class, domestic violence exposure class, and multiple trauma exposure class. More importantly, the primary contribution of this study is that migrant boys and girls differed in terms of their distribution among CTE classes and in the patterns of CTEs.

The proportion of the low trauma exposure class for both boys and girls was the largest, representing 57.2% of males and 64.6% of females. This result was consistent with prior studies showing that boys were likely to endorse more types of victimization [17,42]. The smaller proportion of boys in this class might result from boys’ tendency to engage in more risky activities and seek adventures [43].

Similar to the estimates on large-scale samples in Western countries showing that multiple exposed individuals accounted for a small fraction of the total population (approximately 5%) [21], the multiple trauma exposure class represented 5.4% of boys and 5.3% of girls in our study. The other two multiple-exposure subgroups made up one-third of the male and female sample; more boys fell into classes characterized by vicarious trauma exposure, while more girls fell into the domestic violence exposure class.

There were also morphological differences in the same class of CTEs across sexes. For the multiple trauma exposure class, the pattern in the male data was characterized by a higher likelihood of sexual harassment, experiencing and seeing domestic violence and violence away from home than in the female data. Boys in the domestic violence exposure class were significantly more likely to suffer domestic violence themselves or witness domestic violence than girls in the same class. Similarly, boys in the vicarious trauma exposure class endorsed more domestic violence along with painful medical treatment, while girls in this class were unlikely to be exposed to domestic violence. Overall, boys were more likely to be exposed to trauma than girls within the same CTE class. These findings replicated prior studies identifying males as more likely to experience physical abuse than females during childhood [14,15,25,26,27].

### 4.2. Predictive Factors of CTE Classes

To examine sex differences in predictive factors for CTEs, we compared the role of demographic variables in predicting CTE classes in migrant boys and girls. Parents’ marital status differentiated the class membership of both boys and girls. Compared to parents in their first marriage, girls having remarried parents were more likely to fall into the domestic violence exposure class and vicarious trauma exposure class. However, the differential effect of a remarried family was insignificant for boys. The results suggested that girls in a stepfamily might have a higher risk of witnessing and undergoing domestic violence, possibly due to the strong preference for sons among Chinese rural-to-urban migrants, reflected by the ratio of school-age boys to girls that reached 1.39 to 1 in a survey on migrant children in Beijing [35]. Parents with a son preference were found to discriminate against daughters in distributing sparce resources, including parental care and healthcare [44]. The lack of parental care and low status in the family could make girls vulnerable to direct violence from family members. Meanwhile, their mothers might face discrimination in the family and ridicule in the community due to the daughters she had given birth to [44], placing the couples at higher risk of conflicts and domestic violence. However, boys having divorced parents had a greater probability to be in the domestic violence class compared to those with parents in their first marriage. Single parenthood was shown to be especially stressful and was associated with lower quality parenting that may hamper children’s well-being [45]. For migrant parents with economic burdens, such a status could intensify their stress in raising children. Another possible reason underlying this finding might be that parents from rural areas in China tend to resort to physical abuse when instructing their children, especially their sons. In addition, both boys and girls having divorced parents had a greater possibility to be in the multiple trauma exposure class. Overall, these findings indicated that first-marriage families may provide more adequate parental supervision that results in fewer CTEs for migrant children [46]. Compared to their urban peers, migrant children were more likely to experience more punishment and strictness from their parents and have a relatively low-quality parent–child attachment [5]. Based on this, these changes in family structure were particularly important in determining traumatic experiences in childhood for migrant children [31]. Meanwhile, divorce and remarriage may have sex differences in the risk of domestic violence.

Fathers’ education level predicted different CTE classes among boys and girls. Among the children’s fathers having a primary school education or below, boys had greater odds to fall into the domestic violence exposure class, while girls had greater odds to fall into the multiple trauma exposure class. Fathers with lower levels of education might be a risk factor of becoming perpetrators [47], resulting in more corporal punishment for their children. As boys were found to report more conflict with parents than girls [48], they were more likely the target of domestic violence from fathers with low education levels. This was in line with a previous study on child maltreatment in western China, showing that boys had a higher likelihood of experiencing such violence in underprivileged areas in China [49]. Furthermore, a low education level partly reflected low SES, which could lead to living a community with a higher risk of multiple trauma exposure. Generally, migrant youth reported moderate levels of economic stress and discriminatory abuse [4]. Living in areas of concentrated poverty and disadvantage elevated the risk for migrant children to both be exposed and engaged in violence [50]. Meanwhile, communities characterized by greater violence were likely to exhibit lower education levels and incomes, and those lacking resources often failed to move out [51]. In addition, parenting challenges might be exacerbated by the stress of living in such communities and the lack of social capital [52]. Thus, these reasons might explain why girls with low-educated fathers were more likely to experience vicarious types of CTEs. Therefore, prevention plans on domestic violence can target boys having fathers in low education levels, and interventions might need to cover additional adversities in the communities for girls’ fathers with lower education levels.

Additionally, a low level of peer support was associated with the multiple trauma exposure class for boys and the domestic violence exposure class for girls. A possible reason might be that peer support helps reduce multiple types of CTEs among boys, such as protecting them from community violence and the perception of discrimination, which was more often reported by migrant boys and migrant girls. Boys lacking peer support tend to be directly exposed to discrimination from other peer groups, which was associated with increased behavioral problems [53] along with risky behaviors [54] among boys. These behaviors can result in a higher risk of endorsing substantial CTEs of different types. Moreover, increased support from friends was related to decreased bullying perpetration after 1 year for boys but not for girls [55], leading to reduced CTEs from the community and schools. On the other hand, girls exposed to domestic violence, especially physical violence between their father and mother, were reported to be approximately three times more likely to become bullies compared to girls not exposed to domestic violence, leading to unfavorable peer relationships [56]. In addition, low-income families and disharmonious family relationships were linked with a higher level of difficulty in peer relationships [11]. Their experience of social rejection by urban peers and adults has severe consequences on children’s social functioning, mental well-being and physical health [3]. As migration disturbs their relationships with friends and distant family members, leading to insufficient social support [32], interventions could focus on promoting peer support networks among migrant children.

For both genders, an older age was related to classes with more CTEs. One possible explanation is that those older in age may have undergone a greater number of potentially traumatic events, such as medical treatment or the death of relatives, than those with younger age. Another possibility could be that the migrant children generally became more risk-seeking when they reached adolescence, as it is often viewed as a period featuring immature self-regulation and a tendency to engage in risky behaviors [57], increasing their risk of encountering traumatic events. 

Overall, stratifying the data by sex was important in exploring the different roles of demographic variables among boys and girls. Our findings underscored the significance of providing adequate support for migrant children, which was effective in reducing CTEs and preventing their recurrence [15].

### 4.3. Limitations

The present study still has a few limitations. First, the assessment of childhood CTEs was reliant on retrospective recall, which could imply reporting bias. Second, the cross-sectional design hindered the identification of temporal ordering of CTE classes and predictive factors, limiting the capacity to draw causal conclusions. Third, sexual assault was incompletely documented in the study. The checklist used to examine CTEs only contained sexual harassment as a milder form of sexual abuse. Finally, data on CTE exposure and predictive factors based on rural-to-urban migrant children required further comparison with local child populations in China. However, CTE and its predictors in local children are still unclear in China. Therefore, we call for extended research in exploring childhood trauma of Chinese children.

### 4.4. Implications

The present study was the first to explore sex differences in patterns of childhood trauma and predictive factors of CTE classes using a large-scale sample of Chinese migrant children. It demonstrated that various dimensions of adversity, such as the number, composition and nature of events, should be considered in trauma studies [58]. In addition, the significant sex differences in our findings underscored the need to discover unique patterns of CTEs by separating the data by sex. Moreover, the patterns of CTEs among boys and girls can inform clinical work targeting children’s mental health, highlighting the need to handle the trauma history of boys and girls differently according to their constellations of risk.

With findings on different environmental and personal predictive factors of CTE classes for boys and girls, this research also holds strong implications for gender-specific treatment services and prevention plans for policy makers and clinical practitioners. It highlights the need to develop intervention programs that target their social skills, family relationships, and the community where migrant boys and girls are embedded. This is consistent with a previous study on family migration that proposed potential measures for promoting individual adjustments, conducting family interventions and providing community support to foster the development of migrant children [11]. Psychological guidance is also critical for students in migrant schools.

## 5. Conclusions

In this study, we found that sex differences existed in the shape and proportion of CTE patterns. Specifically, boys had a higher risk of childhood trauma than girls among Chinese rural-to-urban migrant children. Additionally, we identified a series of predictive factors of CTE classes in boys and girls. Gender differences were observed in predicting classes based on several factors, including parental marital status, peer support, and fathers’ education levels. The results highlight sex as a critical consideration in prevention and treatment efforts when dealing with migrant children and indicate the need to better understand protective and risk factors for childhood trauma to promote the healthy development of disadvantaged children.

## Figures and Tables

**Figure 1 children-10-00734-f001:**
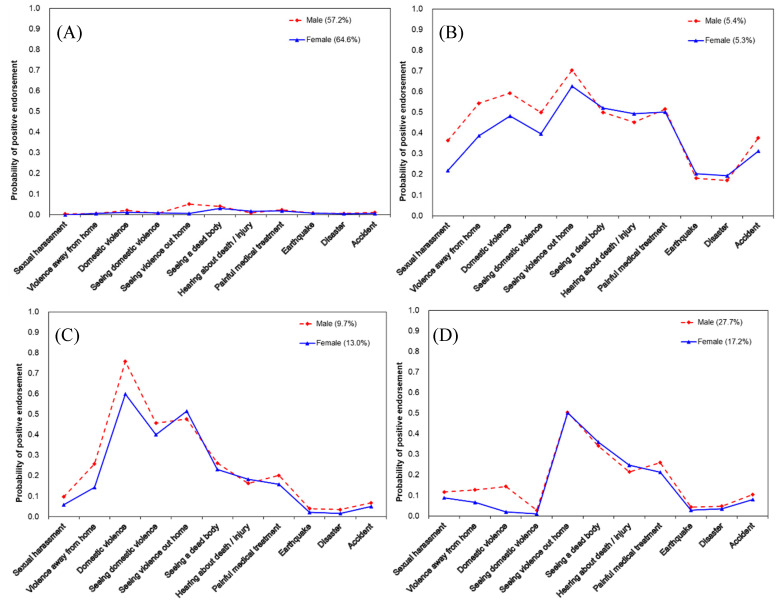
Four classes of childhood trauma in males and females: (**A**) Class 1: Low trauma exposure, (**B**) Class 2: Multiple trauma exposure, (**C**) Class 3: Domestic violence exposure, (**D**) Class 4: Vicarious trauma exposure.

**Table 1 children-10-00734-t001:** Sociodemographic features of the participants (*n* = 16,140).

	Category	*N*	Weighted %
Sex	Male	9247	57.3
	Female	6662	41.3
Age (years)	8–9	2081	12.9
	10	4267	26.4
	11	3678	22.8
	12	2658	16.5
	13–14	2472	15.4
	15–17	818	5.1
Father’s educational background	Primary school or below	4257	26.4
	Junior high school	5860	36.3
	High school or above	3440	24.3
Mother’s educational background	Primary school or below	6355	39.4
	Junior high school	4618	28.6
	High school or above	2077	12.9
Marital status of the parents	First marriage	14,737	91.3
	Divorced	473	2.9
	Remarried	638	4.0
The age of child when father migrated for work (years)	0	2338	14.5
	1–6	5531	34.3
	6–12	1988	12.3
	>12	81	0.5
The age of child when mother migrated for work (years)	0	1885	11.7
	1–6	4889	30.3
	6–12	2442	15.1
	>12	120	0.7
The age when child migrated (years)	0	1941	12.0
	1–6	3977	24.6
	6–12	4652	28.8
	>12	569	3.5

Notes. Due to missing data, the total number of observations for some variables was less than 16,140.

**Table 2 children-10-00734-t002:** Fit indices for two- to five-class solutions of CTE for males (*N* = 9395) and females (*N* = 6745).

Number of Classes	AIC	BIC	a-BIC	LMR-LRT	Proportion of the Smallest Class
Males					
2	60,360.89	60,525.29	60,452.20	<0.001	33.40%
3	60,002.32	60,252.50	60,141.28	<0.001	13.58%
**4**	**59,831.40**	**60,167.40**	**60,018.00**	**0.016**	**5.40%**
5	59,781.09	60,202.82	60,015.33	0.129	0.50%
Females					
2	36,449.18	36,605.96	36,532.87	<0.001	28.08%
3	36,290.60	36,529.18	36,417.96	0.161	12.91%
**4**	**36,165.19**	**36,485.57**	**36,336.22**	**<0.001**	**5.30%**
5	36,157.15	36,559.33	36,371.84	0.506	4.88%

Notes. AIC, BIC, and a-BIC refer to Akaike information criterion, Bayesian information criterion, and Bayesian information criterion adjusted for sample size. LMR–LRT refers to Lo-Mendell–Rubin adjusted likelihood ratio test. Indices of selected models are highlighted in bold type.

**Table 3 children-10-00734-t003:** Demographic variables of males according to classes of CTEs (*N* = 9395).

	Low *n*/Mean (%/*SD*)	Domestic *n*/Mean (%/*SD*)	Vicarious *n*/Mean (%/*SD*)	Multiple *n*/Mean (%/*SD*)	Total *n*	F/χ^2^
Age	11.13 (1.59)	11.56 (1.76)	11.60 (1.77)	11.72 (1.86)		212.94 ***
Family support	2.66 (0.82)	2.45 (0.82)	2.67 (0.84)	2.47 (0.85)		71.43 ***
Peer support	2.66 (0.85)	2.53 (0.87)	2.72 (0.86)	2.53 (0.89)		57.71 ***
Father’s education						
Primary school or below	1440 (58.35)	246 (9.97)	669 (27.11)	113 (4.58)	2468	26.97 ***
Junior high school	1959 (58.90)	243 (7.31)	980 (29.46)	144 (4.33)	3326	
High school or above	1228 (60.43)	126 (6.20)	595 (29.28)	83 (4.08)	2032	
Mother’s education						
Primary school or below	2134 (58.59)	301 (8.26)	1035 (28.42)	172 (4.72)	3642	7.62
Junior high school	1541 (59.02)	191 (7.32)	782 (29.95)	97 (3.72)	2611	
High school or above	731 (60.26)	90 (7.42)	341 (28.11)	51 (4.20)	1213	
Parental marital status						
First marriage	5220 (60.72)	640 (7.44)	2379 (27.67)	358 (4.16)	8597	25.53 ***
Divorced	139 (50.36)	33 (11.96)	81 (29.35)	23 (8.33)	276	
Remarried	188 (56.63)	27 (8.13)	104 (31.33)	13 (3.92)	332	

Notes. *** *p* < 0.001. Low, Low trauma exposure; Domestic, Domestic violence; Vicarious, Vicarious trauma exposure; Multiple, Multiple trauma exposure. Due to missing data, the total count of each variable was less than 9395.

**Table 4 children-10-00734-t004:** Demographic variables of females according to classes of CTEs (*N* = 6745).

	Low *n*/Mean (%/*SD*)	Domestic *n*/Mean (%/*SD*)	Vicarious *n*/Mean (%/*SD*)	Multiple *n*/Mean (%/*SD*)	Total *n*	F/χ^2^
Age	10.92 (1.58)	11.54 (1.86)	11.39 (1.83)	11.84 (2.04)		243.93 ***
Family support	2.64 (0.83)	2.37 (0.82)	2.62 (0.85)	2.43 (0.89)		98.48 ***
Peer support	2.60 (0.81)	2.43 (0.82)	2.61 (0.83)	2.55 (0.84)		41.47 ***
Father’s education						
Primary school or below	1246 (69.65)	251 (14.03)	209 (11.68)	83 (4.64)	1789	24.23 ***
Junior high school	1863 (73.52)	272 (10.73)	308 (12.15)	91 (3.59)	2534	
High school or above	1059 (75.21)	152 (10.80)	160 (11.36)	37 (2.63)	1408	
Mother’s education						
Primary school or below	1957 (72.13)	335 (12.35)	323 (11.91)	98 (3.61)	2713	2.82
Junior high school	1473 (73.39)	230 (11.46)	231 (11.51)	73 (3.64)	2007	
High school or above	641 (74.19)	92 (10.65)	103 (11.92)	28 (3.24)	864	
Parental marital status						
First marriage	4544 (74.01)	683 (11.12)	704 (11.47)	209 (3.40)	6140	38.73 ***
Divorced	129 (65.48)	24 (12.18)	26 (13.20)	18 (9.14)	197	
Remarried	194 (63.40)	55 (17.97)	45 (14.71)	12 (3.92)	306	

Notes. *** *p* < 0.001. Low, Low trauma exposure; Domestic, Domestic violence; Vicarious, Vicarious trauma exposure; Multiple, Multiple trauma exposure. Due to missing data, the total count of each variable was less than 6745.

**Table 5 children-10-00734-t005:** Multinomial logistic regressions for predictors of CTE class membership of males (*N* = 9395).

Predictors	Domestic vs. Low (Ref. = Low)		Vicarious vs. Low (Ref. = Low)		Multiple vs. Low (Ref. = Low)	
	OR	95% CIs	OR	95% CIs	OR	95% CIs
Age	1.15 ***	[1.10, 1.21]	1.18 ***	[1.15, 1.21]	1.22 ***	[1.15, 1.29]
Parental marital status:						
Divorced	1.80 **	[1.22, 2.65]	1.25	[0.95, 1.65]	2.16 ***	[1.37, 3.40]
Remarried	1.05	[0.69, 1.59]	1.16	[0.91, 1.48]	0.89	[0.50, 1.58]
First marriage (Ref.)	–	–	–	–	–	–
Father’s education:						
Primary school or below	1.34 **	[1.08, 1.65]	0.93	[0.82, 1.06]	1.04	[0.79, 1.37]
Junior high school	1.08	[0.87, 1.32]	0.96	[0.85, 1.08]	1.06	[0.82, 1.38]
High school or above (Ref.)	–	–	–	–	–	–
Family support	0.78 ***	[0.70, 0.86]	1.00	[0.94, 1.07]	0.84 **	[0.74, 0.96]
Peer support	0.92	[0.84, 1.02]	1.06	[1.00, 1.12]	0.88 *	[0.77, 0.99]

Notes. * *p* < 0.05; ** *p* < 0.01; *** *p* < 0.001. Low, Low trauma exposure; Domestic, Domestic violence; Vicarious, Vicarious trauma exposure; Multiple, Multiple trauma exposure; OR, Odds ratio; Ref., reference group. To deal with missing data, multiple imputation was utilized, resulting in a sample size of 9395 cases in the imputed data sets, with 1948 cases having missing values for independent variables.

**Table 6 children-10-00734-t006:** Multinomial logistic regressions for predictors of the CTE class membership of females (*N* = 6745).

Predictors	Domestic vs. Low (Ref. = Low)		Vicarious vs. Low (Ref. = Low)		Multiple vs. Low (Ref. = Low)	
	OR	95% CIs	OR	95% CIs	OR	95% CIs
Age	1.21 ***	[1.16, 1.27]	1.18 ***	[1.12, 1.23]	1.31 ***	[1.22, 1.41]
Divorced	1.11	[0.71, 1.74]	1.25	[0.81, 1.92]	2.82 ***	[1.67, 4.74]
Remarried	1.66 **	[1.22, 2.27]	1.42 *	[1.02, 1.98]	1.35	[0.77, 2.37]
First marriage (Ref.)	–	–	–	–	–	–
Father’s education:						
Primary school or below	1.15	[0.94, 1.41]	1.03	[0.84, 1.26]	1.50 *	[1.04, 2.16]
Junior high school	0.88	[0.72, 1.08]	0.95	[0.79, 1.16]	1.21	[0.84, 1.73]
High school or above (Ref.)	–	–	–	–	–	–
Family support	0.76 ***	[0.69, 0.84]	1.01	[0.92, 1.12]	0.82 *	[0.69, 0.98]
Peer support	0.86 **	[0.78, 0.96]	1.00	[0.90, 1.10]	0.99	[0.83, 1.17]

Notes. * *p* < 0.05; ** *p* < 0.01; *** *p* < 0.001. Low, Low trauma exposure; Domestic, Domestic violence; Vicarious, Vicarious trauma exposure; Multiple, Multiple trauma exposure; OR, Odds ratio; Ref., reference group. To deal with missing data, multiple imputation was utilized, resulting in a sample size of 6745 cases in the imputed data sets, with 1261 cases having missing values for independent variables.

## Data Availability

The data presented in this study are available on request from the corresponding author.

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
