# Peer review of "Sex Differences in Patterns of Childhood Traumatic Experiences in Chinese Rural-To-Urban Migrant Children"

_children, 2023, doi:10.3390/children10040734_

Round 1
Reviewer 1 Report
It is a very relevant and interesting study that approaches gender differences from a constructive point of view, from which different intervention options can be derived.
As for the summary, it fulfils the necessary sections and has complete information.
As for the introduction, it would be useful to know why you decided to carry out this research on this particular population, as well as the explanation that there was nothing written on the subject. Did you think that there might be a difference for some social, cultural or psychological reason? Little has been explained about it, only this paragraph:
In line with this, they are often exposed to childhood traumatic experiences (TEs), which are linked to a series of negative mental, physical and behavioural out-comes. For example, they experienced more community violence, punishment from parents, and negative life events than their urban peers.
This explanation would need to be expanded further, as from a non-Chinese context it is not clear why this population group may be disadvantaged.
In terms of data analysis, it would be very interesting to know certain variables: did they live through the migration process or did their parents migrate before the child was born? If they migrated, at what age did they migrate from the village to the city?
The sample has a very wide age range, it would be interesting to explore differences in the sample itself by age subgroup. Are there differences between ages? For example, does the likelihood of trauma increase with increasing age? Perhaps these data could be the subject of a whole other article, but it would be appropriate to mention something about this in the present article.
The conclusion section is appropriate.
Author Response
Response: Thank you for your encouragement and thoughtful review of the manuscript. We are pleased to hear that you found our study to be relevant and interesting.
As for the introduction, it would be useful to know why you decided to carry out this research on this particular population, as well as the explanation that there was nothing written on the subject. Did you think that there might be a difference for some social, cultural or psychological reason? Little has been explained about it, only this paragraph:
In line with this, they are often exposed to childhood traumatic experiences (TEs), which are linked to a series of negative mental, physical and behavioural out-comes. For example, they experienced more community violence, punishment from parents, and negative life events than their urban peers.
This explanation would need to be expanded further, as from a non-Chinese context it is not clear why this population group may be disadvantaged.
Response: Thank you for pointing out this issue with the lack of background information of migrant children. We have added information about the dualistic household registration system and the lack of parental supervision, which may serve as social and psychological reasons for the vulnerability of migrant children.
(p. 1) The dualistic household registration system that assigns the children to either rural or urban status limits their access to social welfare and educational resources, making them vulnerable to discrimination based on their migrant status [49]. Furthermore, the marginalized living conditions experienced by many migrant children also lead to reduced parental supervision, which can result in poor parent-child attachment [45].
In terms of data analysis, it would be very interesting to know certain variables: did they live through the migration process or did their parents migrate before the child was born? If they migrated, at what age did they migrate from the village to the city?
Response: Thank you for pointing out this issue. We have expanded the table of demographics to include the age of children when their father, mother, or themselves migrated. Please see Table 1.
The sample has a very wide age range, it would be interesting to explore differences in the sample itself by age subgroup. Are there differences between ages? For example, does the likelihood of trauma increase with increasing age? Perhaps these data could be the subject of a whole other article, but it would be appropriate to mention something about this in the present article.
Response: Thank you for your valuable advice. We found the likelihood of trauma do increase with increasing age from the results of multiple logistic regression, which indicated that older age was a risk factor, related to classes having more CTE types for both boys and girls. We have also added more possible explanation in the discussion part.
(p. 12) For both genders, an older age was related to classes with more CTEs. One possible explanation is that those older in age may have undergone a greater number of potentially traumatic events, such as medical treatment or the death of relatives, than those with younger age. Another possibility could be that the migrant children generally became more risk-seeking when they reached adolescence, as it is often viewed as a period featuring immature self-regulation and a tendency to engage in risky behaviors [59], increasing their risk of encountering traumatic events.
The conclusion section is appropriate.
Response: Thank you for pointing out this issue. We added conclusion section.
(p. 13) In this study, we found that sex differences existed in the shape and proportion of CTE patterns. Specifically, boys had a higher risk of childhood trauma than girls among Chinese rural-to-urban migrant children. Additionally, we identified a series of predictive factors of CTE classes in boys and girls. Gender differences were observed in predicting classes based on several factors, including parental marital status, peer support, and father's education level. The results highlight gender as a critical consideration in prevention and treatment efforts when dealing with migrant children and indicate the need to better understand protective and risk factors for childhood trauma to promote the healthy development of disadvantaged children.

Reviewer 2 Report
This article was nicely described in the first and subsequent paragraphs, with a compelling description of why this research was important and necessary (e.g., previous research only completed with Western samples).
I found the significant use of TE, especially, confusing. Maybe it could be CTE (Childhood Trauma Experiences) rather than TE, which might make paragraphs with your acronym more readable. For example, para. 2 includes TE three times in the first sentence, which makes it more difficult to follow your point (8 times in the paragraph, plus two uses of LCA). Maybe there are some places where using some other phrase would be helpful.
This paper stands alone, as it points out that these are issues for rural-urban migrant children, although I would like to see comparisons with local children. I hope such research is in the future.
I appreciated the authors' descriptions of why males and females might have different predictors of group membership. This should probably be further emphasized as tentative (e.g., "They might," "It is possible…").
Author Response
Response: Thank you for your words of encouragement and careful reading of the manuscript. We will improve this study on Chinese migrant children based on your suggestions, and we hope that our study can serve as a valuable reference for future research on comparisons with local children.
I found the significant use of TE, especially, confusing. Maybe it could be CTE (Childhood Trauma Experiences) rather than TE, which might make paragraphs with your acronym more readable. For example, para. 2 includes TE three times in the first sentence, which makes it more difficult to follow your point (8 times in the paragraph, plus two uses of LCA). Maybe there are some places where using some other phrase would be helpful.
Response: Thanks for your great suggestion on improving the accessibility of our manuscript. We agree that it would be more specific with CTEs instead TEs. We have replaced most of TEs with CTEs.
I appreciated the authors' descriptions of why males and females might have different predictors of group membership. This should probably be further emphasized as tentative (e.g., "They might," "It is possible…").
Response: Thank you for your valuable advice. We have used more tentative expressions when describing the gender differences in predictors of group membership.
(e.g., p. 11) Furthermore, a low education level partly reflected low SES, which could lead to living a community with a higher risk of multiple trauma exposure.
(e.g., p. 12) Thus, these reasons might explain why girls with low-educated fathers were more likely to experience vicarious types of CTEs.
Reviewer 3 Report
This paper looks comparatively at rural-to-urban migrant children within China and their mental health vulnerability. Much research focuses on international migration, but internal migration is equally important, especially within China. It is also essential to study the mental health of minors moving across distances. This paper points at differences across gender and adds to what should be a growing research area.
Author Response
Thank you for providing us with this great opportunity to submit a revised version of our manuscript entitled “Sex Differences in Patterns of Childhood Trauma among Chinese Rural-to-Urban Migrant Children” (Manuscript Number: ijerph-2229191). We are greatly encouraged by your recognition of this study on internal migration within China, and the traumatic experience of migrant children. We hope that you will find this revised version satisfactory, and look forward to hearing from you.